# Using systems-mapping to address Adverse Childhood Experiences (ACEs) and trauma: A qualitative study of stakeholder experiences

Thi Hoang Vu[1]*, Jared Bishop[2], Leigh McGill[2], Luke Valmadrid[2], Shelley Golden[2], Dane Emmerling[2], Seth Saeugling[3]

1 Department of Social and Behavioral Sciences, Yale School of Public Health, Yale University, New Haven, Connecticut, United States of America, 2 Department of Health Behavior, Gillings School of Global Public Health, University of North Carolina, Chapel Hill, North Carolina, United States of America, 3 Rural Opportunity Institute (ROI), Housed at Area L AHEC, Rocky Mount, North Carolina, United States of America

* thi.vu@yale.edu

**Data Availability Statement:** All relevant data are within the manuscript and its Supporting information files.

## Abstract

Adverse childhood experiences (ACEs) and trauma have been linked to decreased psychosocial and physiological health functioning. While various individual and community-level interventions to address ACEs have been reported, one novel approach that has not been explored in detail is a community-engaged causal loop diagramming project, or systems mapping project (SMP), in which diverse stakeholders work together to document the forces that are creating the outcomes and patterns within the community. To better document and understand the impact of participation in an SMP, we conducted in-depth, qualitative interviews with 16 stakeholders who were involved in a systems-mapping process facilitated by a local nonprofit in Eastern North Carolina. We used an iterative, content analysis coding process to generate and analyze themes from these interviews. Three major themes emerged: 1) Recognition and understanding of own trauma, 2) Trauma as both a community issue and an individual issue, and 3) Systems-mapping as a conceptual tool with practical benefits. All participants strongly recommended the systems-mapping approach to other communities and believed that it is a valuable tool for empowerment and provided several considerations for future organizers of similar systems-mapping projects. Our findings suggest that systems mapping is a feasible, transferable, and promising modality for understanding and addressing ACEs at the individual, interpersonal, and community-levels, as well as for putting community voices at the forefront of efforts to address ACEs.

## Introduction

Adverse childhood experiences (ACEs) are stressful, traumatic events experienced by children that can result in substantial emotional and chronic stress, and continue to impact their lives as adults. Some examples of ACEs include neglect, abuse, experiencing discrimination, bullying, and witnessing events like maltreatment of family members and community violence [1]. Data from 23 states in the 2014 Behavioral Risk Factor Surveillance System survey reported that 62% of 214,157 survey participants had experienced at least 1 ACE and 25% reported

**Funding:** SS was a consultant for the Rural Opportunity Institute (ROI) project which led this community-based systems mapping process that was funded by The Anonymous Trust. The funder had no role in any way in the study design, data collection and analysis, decision to publish, or preparation of the manuscript. The funder does not have a website, but here is the ProPublica link about the funder (https://projects.propublica.org/nonprofits/organizations/266220561).

**Competing interests:** The authors have declared that no competing interests exist.

having experienced 3 or more ACEs, defined by the survey as incidences of physical, sexual, or emotional abuse, as well as household mental illness, household substance use, incarcerated household member, parental separation or divorce, and household domestic violence, before the age of 18 [2]. Communities of color and lower-income communities face higher risks for experiencing ACEs compared to white and higher income populations [3]. Additionally, a 2018 report from the National Advisory Committee on Rural Health and Human Services suggest that nearly 29% of children living in rural areas experience two or more ACEs compared to 21% of children living in urban areas, and that rural children were more likely to experience abuse and neglect compared to urban children [4].

ACEs and chronic trauma exposure in childhood can result in both immediate and long-lasting health outcomes [5, 6], including increased risk of lung cancer, diabetes, cardiovascular disease, and obesity; mental health conditions like PTSD and depression; and changes to brain structure [7]. Additionally, health-risk behaviors with strong social and environmental determinants, like tobacco use and substance abuse [6, 8–11] have also been linked to ACEs and chronic trauma exposure. Furthermore, higher numbers of ACEs have been associated with less health care use and access [12]. Drawing on studies between 1998 and 2017 in North America, the estimated total healthcare costs of diseases associated with ACEs, such as anxiety, depression, cardiovascular disease, cancer, diabetes, and respiratory disease was $748 billion [13]. Due to the compounding nature of both the morbidities mentioned and health care utilization, persons with ACEs face multiplicative, intersecting barriers to achieving their optimal health [14].

ACEs are heavily shaped by social, historical, and environmental determinants. Thus, the last few decades of literature surrounding ACEs and traumatic stress have highlighted the importance of systemic approaches to address and prevent trauma across multiple societal sectors [10]. However, many primary and secondary prevention efforts are still focused on individuals and families [10]. A systematic review of interventions to improve outcomes for persons who have experienced ACEs found that the most common types of interventions represented in the literature include cognitive-behavioral therapy, motivational interviewing, family therapy, and parent/guardian training to improve mental resilience [15]. Even though people exposed to ACEs have diverse and complex needs beyond the individual-level, community-level interventions that consider broader environmental stressors were sparsely represented [15]. Moving beyond trauma-specific services to trauma-informed systems could not only reduce the negative consequences of trauma and promote healing for individuals [16], but also diversify the evidence base for development of future programs and policies [15]. One promising modality of creating trauma-informed systems is community-based system thinking processes.

In public health, systems thinking views each element that impact individual and community health as interrelated and dynamic [17] and involves looking at how the relationships between individual factors and larger structural and social forces influence health outcomes at the individual and population level [18]. Therefore, systems thinking allows stakeholders and organizations connect upstream and downstream factors specific to a community and identify critical points for interventions across multiple societal levels [19, 20]. One specific tool employed in systems thinking is the creation of a causal loop diagram, more commonly known as a systems map, of feedback loops that illustrate the patterns of forces that are causing and influencing observable outcomes [21]. Relationships between each element in a diagram are represented through arrows. Positive relationships between elements where both elements progress in the same direction are represented through "pluses," and negative relationships between elements where each element progress in opposite directions are represented through "minuses" next to the arrows [21]. Systems thinking, specifically systems mapping, has

provided direction and novel insights for actions taken by community organizations and researchers [22, 23] and have also significantly influenced program implementation and resource allocation of state-level and county-level health initiatives [24, 25]. In particular to ACEs, however, systems thinking has mainly been used to identify potential leverage points for interventions [26, 27]. For example, causal loop diagrams have been developed by researchers to understand how community programs, social services, and the built environment interact to promote social and emotional well-being in children [26] as well as how parental opioid use can be predictors of child maltreatment and children's' maladaptive coping behaviors, thereby perpetuating the cycle of ACEs [27].

In practice, systems-mapping usually happens in an academic setting where the participants doing the mapping are often professional and privileged individuals, such as professors, funders, and researchers [28, 29]. Systems thinking projects are rarely developed through co-creation with local community leaders and community members, especially in contexts such as the rural southern United States [30]. There are few examples that document the impact of systems-mapping within the context of a grassroots community and alongside under-resourced community members with lived experience and proximity to the challenges that are being mapped [28].

Thus, this paper describes a collaborative, community-led systems-mapping project (SMP) facilitated by Rural Opportunity Institute (ROI) and the sub-project (qualitative interviews) of analyzing the impact of the mapping efforts on community members. We seek to add knowledge and examples of how this process can be used with and alongside community members, and how lived experience and insights can drive the mapping process to inform interventions. We engaged ROI stakeholders who participated in the SMP, or who have worked with ROI on initiatives informed by the SMP. The aim of this study is to better understand: 1) individual and community-level impacts of systems-mapping; 2) how systems thinking can be leveraged to address ACEs and trauma; and 3) lessons-learned and recommendations for communities looking to apply systems thinking to addressing health issues.

## Methods

### Context: The ROI systems-mapping process

ROI is a non-profit operating in Edgecombe County, North Carolina, USA. Edgecombe County is a rural county in eastern North Carolina with a population of almost 52,000 in 2019 [31]. The median household income in 2019 was approximately $36,000. About 21% of the population live below the poverty line [31]. The overall objective of ROI's systems science work is to increase the capacity of public agencies within Edgecombe County to more effectively address the trauma/ACEs faced by members of the community and end generational cycles of trauma and poverty. Through building networks, strengthening capacity, and supporting existing programs and public agencies within the system to implement best practices around becoming trauma-informed, ROI aims to support a holistic community effort where the dominant response across public agencies within the system is to provide healing, restorative practices, and skill building as a response to trauma [32].

Between September 2017 and May 2018, ROI hosted eight community meetings to create the systems-map (Fig 1). The meetings happened every eight weeks and were held at rotating locations in trusted spaces in the community, such as a local recreation center, a public school, the community college, the county government auditorium, and a local business. Meetings were open to the public and engaged 413 community members in total (an average of 52 members per meeting) [33]. Demographic information for these stakeholders can be found in S1 Table. At these meetings, community members discussed forces that contribute to the current

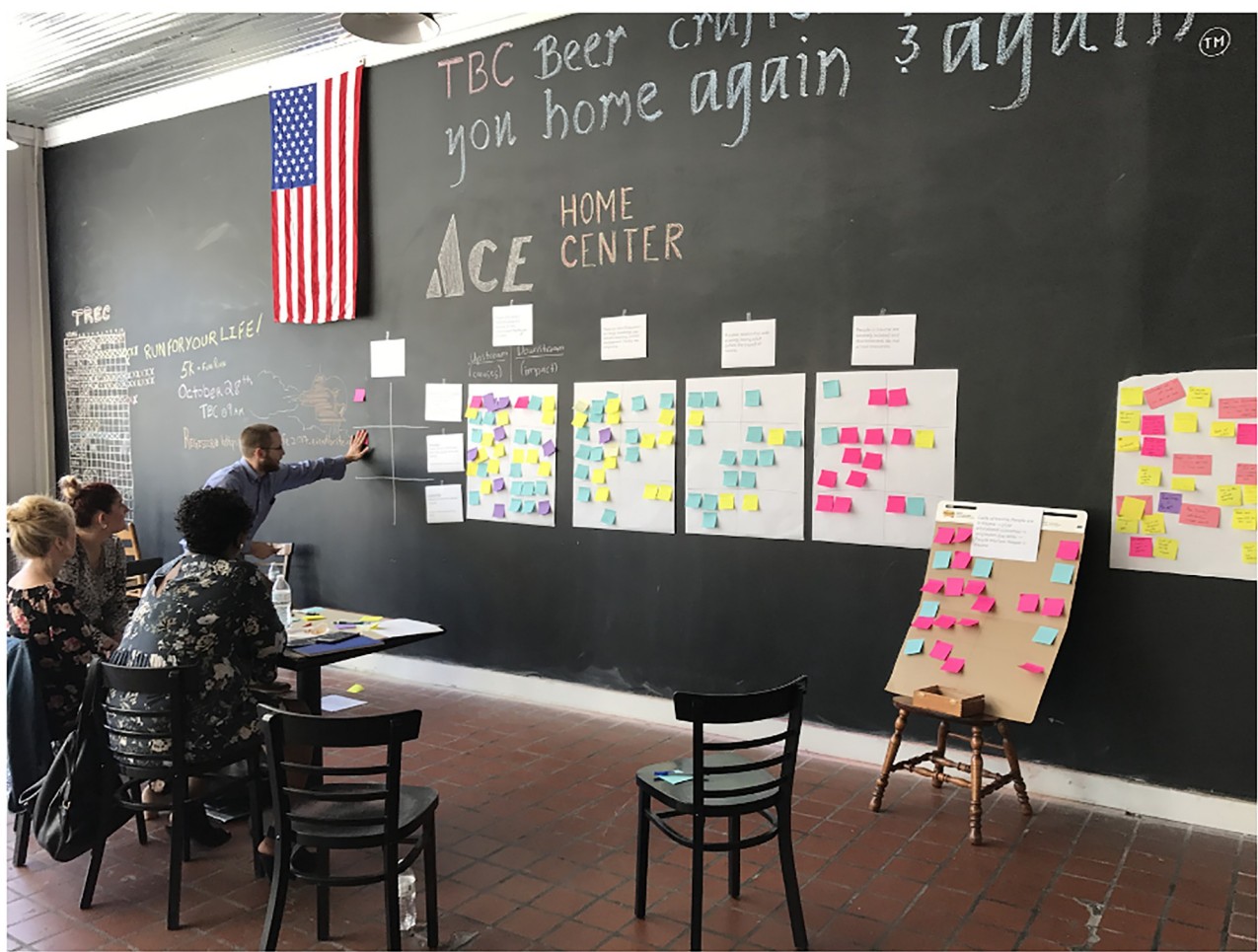

**Fig 1. A meeting engaging community members to brainstorm ideas for the systems-map.**

level of ACEs/trauma in Edgecombe County, as well as forces that help provide healing. First, community members shared their experiences of where trauma and stress show up in their lives and in the community, along with what spaces, programs, organizations, and people support healing and recovery. For each theme identified in the first step, for example, "high rates of teen pregnancy in our community," community members then brainstormed upstream causes and downstream impacts. Using notecards, sticky notes, and poster paper, community members built a rough draft of the systems map to visualize the interactions between the themes, causes, and impacts identified [33]. Using the rough draft built by stakeholders (Fig 2), ROI then contracted with Engaging Inquiry, a purpose-built consulting practice that supports communities to use participatory systems mapping [34], to create the final map (Fig 3). ROI and Engaging Inquiry then facilitated a Leverage Workshop with cross-sector community leaders to identify high-leverage intervention points within the map [33].

The final outcome of the map and workshop was the creation of a three-part strategy to address trauma defined by community members: 1) Learn: increase knowledge and skills about what trauma/resilience is, and how to best manage stress; 2) Heal: shift practices and policies away from the currently dominant punitive response and towards a more restorative approach that helps people build skills; and 3) Connect: reconnect youth and adults with

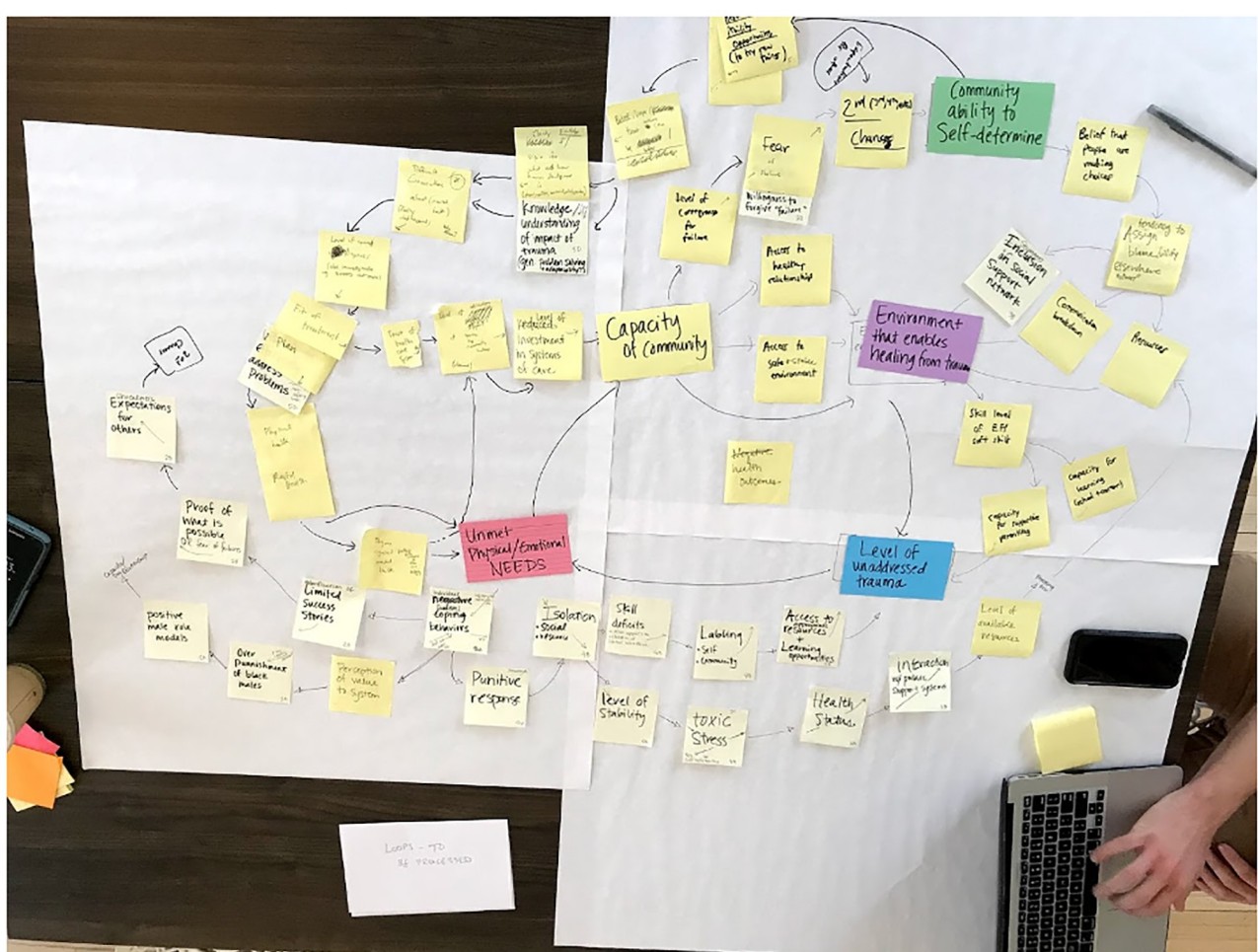

**Fig 2. A rough draft of a working systems-map.**

educational and professional opportunities [33]. ROI then formed a training collaborative of about 20 community members to research evidence-based practices that align with these strategies. To do so, the training collaborative spent about 3 months meeting with 5–8 other communities across the USA that were also conducting trauma and resilience work to learn more about their practices and how they can be applied to Edgecombe County [33]. The systems mapping process guided the development of several initiatives, such as: 1) a local training collaborative that has reached over 13,000 individuals through programs such as evidence-informed Reconnect for Resilience Trainings, listening circles, and awareness-building presentations; 2) a community accountability board made up of residents who oversee the training work; and 3) a biofeedback breathing program using HeartMath technology implemented in a local detention center and middle school [35].

## Study design

This study adopted a qualitative design consisting of semi-structured, one-on-one interviews with ROI stakeholders to answer the following research questions: 1) What were the perceived individual, interpersonal, and societal level impacts of the SMP? and 2) How can systems-thinking be leveraged in other communities to address ACES and other public health issues?

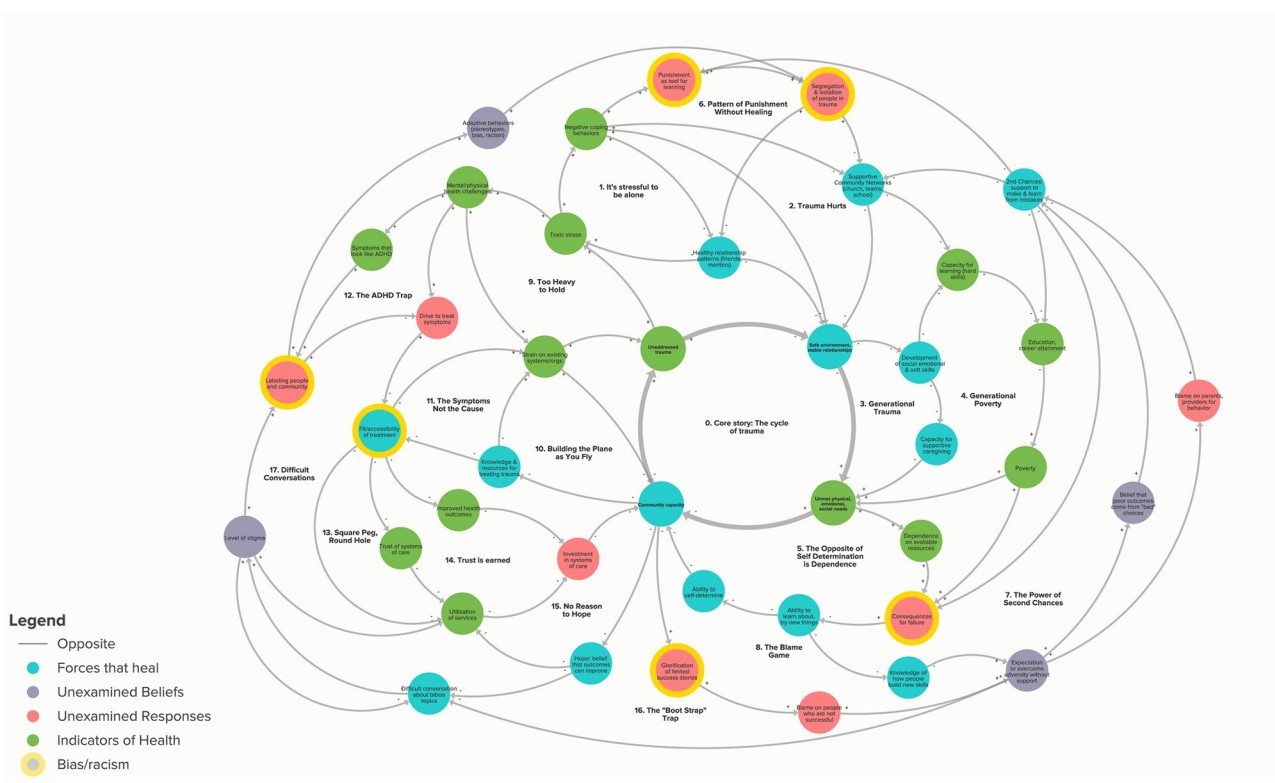

**Fig 3. The final systems-map.**

ROI collaborated with a group of public health graduate students (THV, JB, LM, and LV) with field experience and didactic training in qualitative methods at a public research university in North Carolina to design and implement this study. This study was reviewed prior to the start of participant recruitment and deemed exempt by the University of North Carolina Institutional Review Board (#20–1983). This study's methods and findings are reported following COREQ guidelines for qualitative research [36].

## Sampling

We used purposive sampling and recruited participants electronically by posting an announcement in ROI's monthly email newsletter, distributing electronic fliers, and making social media posts. These efforts reached approximately 200 individuals. Interested individuals then contacted the students determine eligibility and schedule an interview. Participants were eligible if they either participated in any activity of the SMP, and/or were involved in initiatives informed by the SMP.

## Interview guides

The students (THV, JB, LM, and LV) conducted semi-structured, one-on-one interviews with current and former ROI stakeholders. All interviewers followed two semi-structured interview guides with questions and probes developed collaboratively between ROI staff (SS), faculty (SG), and graduate students (THV, JB, LM, LV, DE) at the university. We used the social-ecological framework to develop the interview guides in order to capture the interplay between individual, interpersonal, community, and societal impacts of the SMP [37]. Those

who participated in initiatives informed by the SMP were asked about the personal and inter-personal impacts of the SMP, community-level impacts of the SMP, and recommendations and advice for other communities looking to conduct a similar project. Those who were directly involved in the SMP were also specifically asked about their individual experiences and involvement with creating the map itself. Each interview session consisted of only one interviewer and one participant, and the interviewers had no prior relationship or interactions with any of the participants. Participants provided verbal consent and received a $30 electronic gift card for completing the interview. Interview guides can be found in the S2 Table.

## Data collection and analysis

Interviews were conducted through Zoom (version 5.4.8), a video and voice conferencing plat-form [38]. Interviews were conducted over video call *(n = 11)*, and voice call *(n = 5)*. All inter-views were transcribed by Zoom, and then cleaned and checked for accuracy by the interviewer. Transcripts were uploaded into Dedoose version 8.3.45, 2020 [39], a qualitative coding software, for analysis. We performed content analysis [40] using a thematic approach [41]. In the deductive coding phase, THV, JB, LM, and LV (i.e., the coding team) developed a preliminary codebook based on topics covered in the interview guide. All coders then reviewed a sample of the same two transcripts to re-familiarize themselves with the data. Next, each coder independently coded the sample transcripts to apply preliminary codes and develop emerging codes in the inductive phase. All coders then met to discuss coding application and emergent codes. All coders worked together to refine the codebook and approve a final code-book. Coders were then split into two teams of two (THV and JB; LM and LV). The tran-scripts, including ones used as samples, were divided among the two teams. Within each team, coders then independently coded each transcript and met with each other upon coding com-pletion to resolve any discrepancies in the coding application and make changes to the code-book as necessary. Transcripts were re-coded as needed after coders came to an agreement. Thus, each transcript was coded and reviewed for discrepancies by at least two coders. This process was followed for all transcripts to ensure strong inter-coder reliability. We generated code reports that indicated where and how each code was applied across interviews and used these reports to create initial themes. Coders refined and cross-checked themes with each other and then discussed with the larger research team to ensure consensus. No new codes and/or themes emerged after about half of the interviews were coded, but we continued with the coding process for the remaining transcripts to ensure data saturation, as recommended in the qualitative literature [42–44]. The entire study team (all authors) approved of the final list of themes and illustrative quotes.

## Results

### Participant characteristics

The student team interviewed stakeholders who directly participated in the SMP (n = 8) and stakeholders who were involved in initiatives informed by the SMP (n = 8). The average age for all participants was 53.6 (range 32–73). 47% of participants identified as Black, 60% identi-fied as female, and all participants had completed high school. Additional participant charac-teristics are found in Table 1.

### Themes

The analysis resulted in three major themes: 1) The SMP helped participants better understand their own trauma; 2) The SMP fostered greater interpersonal connections in the community

**Table 1. Self-reported characteristics of interview participants.**

| Characteristic | Overall (n = 16)[a] | Participated in SMP (n = 8)[b] | Did Not Participate in SMP (n = 8) |
|---|---|---|---|
| Mean age (range), years | 53.6 (32–73) | 54.7 (32–73) | 52.6 (45–61) |
| Racial Identity | | | |
| Black | 7 | 3 | 4 |
| White | 8 | 4 | 4 |
| Gender | | | |
| Female | 9 | 1 | 8 |
| Male | 6 | 6 | 0 |
| Highest Level of Education | | | |
| High School | 1 | 0 | 1 |
| Some College | 1 | 1 | 0 |
| Bachelor's | 4 | 2 | 2 |
| Master's | 7 | 2 | 5 |
| Doctorate or more | 2 | 2 | 0 |

[a]One participant's demographic information was not obtained. Percentages were calculated excluding that missing information.

[b]One participant declined to share their age. Values were calculated excluding that missing information.

and allowed participants to view trauma as both a community issue and individual issue; and 3) Participants viewed the SMP as useful conceptual and practical tool. Participants also offered suggestions for other communities looking to conduct a similar SMP. These include ensuring a diversity of stakeholders, letting community members play a leading role in design and implementation, and the importance of genuine relationship-building between non-profit organizers and community members.

## Theme 1: Participants felt a greater recognition and understanding of their own trauma

All participants expressed that the SMP helped them better understand trauma broadly, as well as how it has manifested in their personal lives, by highlighting instances where they shared aspects of their own identities, upbringing, and past traumatic experiences. For example, one participant shared how the SMP was an introspective process that prompted them to recognize their own trauma:

*. . .even as an older adult, you know, you still deal with these traumas that may be buried, you know; and so, you know, you know, a lot of things came back and having to deal with those things or rethink, you know. . .*

(Higher education representative)

Stakeholders expressed that recognition of trauma is one of the first steps in addressing the effects of trauma. One participant, for example, spoke about how the perception that trauma is "just a way of life" masks the effects of traumatic experiences and hinders efforts to address them:

*Something would be diagnosed as trauma, but growing up, it was just a way of life. I mean it, this is what it is. So you don't see it being something that's trauma or whatever because your friends are going through the same thing you were going through. So this is like a way of life*

*versus, you know, something that shouldn't have to take place, you know? And I think that process kind of opened some eyes in the room on what adverse childhood trauma is.*

(Nonprofit employee)

## Theme 2: Participants came to see trauma as both a community issue and an individual issue

Participants reported greater connection to other community members through their participation in the SMP. All participants highlighted how their involvement in the SMP put them in contact with individuals they would have never been connected to otherwise, especially individuals from different sectors, organizations, and backgrounds:

*So there are things, for example, like relationships built for community members that would have never spoken before, so I as a principal wouldn't necessarily have navigated. But now all of a sudden I have these connections to new people. So we're able to build partnerships for support I didn't know existed.*

(Principal)

The SMP also facilitated a discussion about ACEs that took away individual blame on any one person or agency, to instead create a shared understanding around systemic causes of trauma and foster a greater sense of compassion and empathy. In short, participants experienced a shift to a mindset that was less focused on punishment and control, and more focused on healing, skill-building, and restoration both with themselves and with their neighbors.

*I think that we need to make sure that people understand that it can happen to everyone, that, it's not somebody else's problem. It's the community's problem. And lots of people have experienced trauma, but they've kept it to themselves. And they, I mean, that's proven. . .that a lot of people have experienced trauma and just never discussed it. . .but just help people understand that there's nothing to be ashamed of, that what they've experienced is not their fault. Yeah, removing the blame.*

(Community college staff member)

## Theme 3: Participants viewed systems-mapping as a conceptual tool with practical benefits

Stakeholders highlighted that the visualization aspect of systems thinking, specifically systems-mapping, is a unique asset that shines a spotlight on individual, interpersonal, and structural causes of trauma. Systems thinking allowed participants to translate their stories into visual feedback loops, and then to see how these stories and loops connected with each other. Visualizing these stories allowed participants to take a step back and reflect on how their own personal experiences with trauma relate to larger systemic and intergenerational causes of trauma. Not only could participants see their experiences represented in the systems-map, but they also observed interconnections with the experiences of their neighbors.

*If you look at the map itself, there are these areas that have like plus signs and minus signs. And that's kind of like, you know, the areas that the loop is reinforced or weakened, and I thought that it was just really interesting to think about. It's just life right? That like right*

*there are some things compounded that are good and some things that are compounded that are bad. And let's make meaning of the outcome. And I just thought that was really interesting reflection, because again I hadn't considered that before. . .This is like super interesting to see all those circles and cycles working in tandem to create the site problem and or solution. . .*

(Principal)

There are also practical benefits of the SMP. For example, active participation of community members in development and creation of the SMP generated community investment and ownership in the initiatives that arose from the findings of the project. As one participant stated, "*people start gaining interest and they start gaining a voice and by having that voice, they have ownership to what's being done with being said, which is going to increase their commitment. So I don't think they would have the commitment, the level of buy-in had they not done that mapping process*" (Counselor and Licensed Therapist).

Additionally, five participants spoke about how the knowledge gained from the SMP affected how they address trauma in their everyday work. One participant, for example, said that how they approach students as an educator has changed as a result of their participation:

*We want students to understand that we know that you go through things as well. . .when you come to school, 'how can we make school life better, even though you have these things going on?' And building that relationship with the students in order to allow them to be the best student as a whole, instead of just their academics. 'How is home?' You know, 'is there anything that I can do to help you to help make things better for you at home,' as opposed to just come into school learning math, science, social studies and language arts.*

(Guidance Counselor)

Another participant who worked in healthcare noted an increased confidence working with clients on issues surrounding trauma as a result of the SMP:

*I'm learning new things and. . .I would say it's given me some renewed confidence and working with children and adults that have been impacted by trauma.*

(Health Care Provider)

## Considerations for organizers of future systems-mapping projects

Stakeholders who directly participated in the SMP (n = 8) were asked specific questions about their experience with the process itself. All participants emphasized that ROI organizers played an important role in how they perceived the SMP. Participants suggested having dedicated leaders and organizers who are genuine, compassionate, and intentional in their efforts is crucial to stakeholder experience and successful project outcomes. Participants repeatedly emphasized their positive experiences with ROI leadership as one of the reasons they started working with and are continuing to do work around trauma and resilience in partnership with the organization. In particular, participants stressed that it is not enough for organizers to simply include the community; they must also let community stakeholders play a leading role. This includes organizers being open to receiving feedback and making changes as necessary:

*They were very open to feedback and because with the feedback they could improve on what they would do if we gave them honest feedback and the leadership were always open to all,*

*sometimes even criticism about the program. And they were always willing to fix whatever they thought might be broken.*

(Community volunteer)

In addition, all participants expressed that organizers made genuine efforts to connect with community members on a deeper level, both within and outside of the SMP. By making themselves fully available to the community and fully invested in the community, participants felt that organizers were able to form trusting relationships with community members. These efforts created a space that was safe and affirming for community members to share their experiences with trauma.

*They met with everybody you know, and you know you meet with people you let them know you're, you're not here to tear down, you're here to build them up, you're here to participate. Not, you know, observe and you want to pull people together not, you know, pull them apart, you want to, you want to be a part of the solution, not a part of the problem you, you want to help, not hurt. You know, so all those things. I think they did that and they took their time and they listen, listen to everybody.*

(Higher education representative)

Lastly, all participants highlighted that the diversity of voices included in the SMP contributed to its success, noting the varied set of experiences and knowledge that each stakeholder brought to the discussion were extremely valuable. Three participants specifically attributed the diverse representation to the efforts of SMP organizers to invite and include everyone who wanted to take part in the SMP. Stakeholders found it extremely valuable to be able to look at an issue from different perspectives and hear directly from those who have experiences different from their own.

*. . .it was very interesting to hear from healthcare professionals or probation officers, or clergymen and to be talking about the same area and the same issues, but I thought through the lens of not only how they impact events, but what they were doing about it. And so it was the first time that I saw that like, the issue that I see can be solved in more than one way. And I think that that gave me a lot of hope.*

(Principal)

While all participants expressed that the SMP facilitated by ROI was an extremely valuable process, they also noted several areas for improvements that organizers should consider when implementing a similar systems-mapping project. First, despite efforts to include diverse perspectives in the SMP, seven out of the eight participants we interviewed who directly participated in the mapping process itself felt there were still voices that were missing from the discussion. In particular, participants noted that representation from government officials, youth, and residents of neighboring counties was lacking.

Second, participation in the systems-mapping process prompted stakeholders to share personal identities and experiences that were re-traumatizing for some. One participant, for example, mentioned how participating in the SMP prompted him to talk about the loss of loved ones and revisit potential sources of trauma. In addition to emotional investment, participants also emphasized that there is no "quick fix" to systemic and intergenerational trauma, and communities should recognize that the systems mapping approach also requires substantial time and energy:

*Again feet on the ground, hands to the handle. It's just going to take that sort of thing. . .it has taken generations to break and be broken, and it's going to take generations to fix. We got to have people who are willing to change, willing to fix things, willing to self-disclose, willing to self-examine, and, and so yes it'll take a lot of work in every community.*

(Director of Student Support Services)

A complete list of illustrative quotes can be found in S3 Table.

## Discussion

In this study, we documented the stakeholder perceptions, experiences, and impact of participating in an SMP to address ACEs in rural county in Eastern North Carolina. All participants expressed that participation in the SMP generated individual, interpersonal, and community-level benefits, and they would strongly recommend a similar project in other communities looking to address complex health problems. Overall, these findings suggest that systems mapping is a viable and transferable approach with potential to address ACEs at multiple levels by engaging community voices to inform actionable solutions.

Our findings have several important implications. First, while various interventions to address ACEs and trauma have been described in the literature such as psychosocial and behavioral training [45–47]; educational programs [48–50], and arts-based therapies [51–53], there is a dearth of information on the impact of community-engaged interventions, specifically a systems-thinking approach. Yet, there is already some preliminary evidence for the potential of systems-thinking to address public health issues. For example, one study engaged members of a rural Australian community to create a systems-map of determinants of childhood obesity to inform future prevention strategies at the community and policy level [54]. Another study from England engaged staff at various public health institutions to create a systems-map of mental health influences that could help further improve organizational approaches to improve population mental health [55]. Our study adds to the existing literature by illustrating that systems-mapping is a feasible and effective method to address another complex health issue—ACEs and trauma. In addition, our findings add to the evidence-base for how to leverage systems-mapping in a rural, Southern United States context.

Second, the available literature on systems-mapping projects in public health has called for projects to engage a wider, more diverse range of stakeholders. Participants of one project to promote physical activity in the United Kingdom noted that insufficient engagement across the full range of sectors and stakeholders created "gaps" in the map. While the process and final map they created were insightful, there were still missing pieces from limited community engagement that restricted the strength and credibility of the findings [56]. Similar recommendations have been put forth in other studies of systems-mapping projects [54, 55]. In fact, one of the limitations cited by Smith et al.'s causal loop diagramming project focused on parental opioid use and children's emotional and mental wellbeing is that greater engagement of stakeholders with lived experiences would have strengthened the map's effectiveness as a tool to identify intervention points [27]. Friel et al., [57] also noted that engaging only specific groups in the community, such as experts and professionals, may create biased mapping results. Our project expands on these recommendations by engaging many members of the community, all with different backgrounds and experiences, with a particular focus on grassroots community members who do not have affiliations in local formal organizations like the school system, health department, or law enforcement agency. Stakeholders we interviewed expressed that participation of people from so many facets of the community further illuminated the multiple systemic factors underpinning ACEs, as well as connections between systems that perpetuate

trauma within communities. Despite this fact, several stakeholders felt that there were still missing voices. Thus, organizers of future systems-mapping projects should prioritize participant recruitment and engagement early in the process and continuously evaluate their engagement efforts throughout.

Third, there has been a call for systems-thinking projects to provide evidence of "real-world" benefits [58]. Systems-mapping projects pertaining to trauma and ACEs often do not report on the next steps, i.e., the programs and initiatives that come out of the project [26, 27, 29]. Participants in our study felt that diverse representation and inclusion in the SMP created community investment in the process and outcomes of the effort to foster tangible, coordinated, and community-led actions addressing ACEs. At the community-level, participants expressed that the process of creating the map and the map itself resulted in visible changes in how the community responds to trauma. First, findings from the training collaborative have informed programs such as Reconnect for Resilience Trainings, listening circles, and awareness-building presentations that have reached over 13,000 individuals in the community [35]. Residents also formed a community advisory board that oversaw education and training initiatives. To date, over 15,000 people have been trained in what trauma-informed practices look like and how stress impacts the brain and body [35]. Additionally, the systems-mapping process helped support a biofeedback breathing and meditation program implemented in a local detention center and middle school to alleviate anxiety symptoms among participants, as well as the hiring of clinical social workers in schools to work with students who have been expelled or suspended [35]. At the individual and interpersonal levels, participants saw changes in themselves and how they relate to others in their community due to the SMP and the programs resulting from it. Several participants mentioned that they have greater confidence talking about sensitive topics in their line of work, such as being a counselor and educator. All participants felt that without the project and the subsequent programs it informed, they would have never been connected to so many of their fellow community members. Participants described an increased sense of belonging and purpose through these enhanced connections. Our results are in accordance with existing research illustrating that stakeholder engagement can significantly improve the effectiveness of programs by building on existing community efforts and resources [54, 59].

Another powerful takeaway of the individual-level impact is that physically mapping out the feedback loops helped participants who were directly involved in creating the map have a better understanding of ACEs and their own experiences with trauma, even prior to implementation of any programmatic efforts mentioned above. This suggests that the SMP can be an intervention within itself. For example, participants expressed that the act of mapping the loops allowed them to see the formerly invisible forces more explicitly, thereby helping to remove individual blame and a punitive mindset in understanding their own trauma and ACEs. A scoping review of how causal-loop diagramming has been used in public health found that the majority of studies used causal-loop diagramming to inform policy and practice, identify intervention points, and illustrate system complexity [29]. While the review found that there is a wide scope of public health issues represented, such as obesity, community violence, and infectious diseases, only three of the 23 studies captured by the review intended to use systems-mapping to help improve participants' understanding of the public health issue [29]. These results, combined with our study's findings, suggest that there is untapped opportunity to use the act of systems-mapping as an intervention itself to improve individual awareness and understanding of various public health concerns.

Lastly, our findings highlight that it is not enough for systems- mapping efforts to simply include the community, they must also engage the community as leaders and facilitators of these efforts. In doing so, systems-mapping projects can also more accurately capture the

health needs and concerns of marginalized populations to more effectively reduce, racial, ethnic and class disparities in health [58]. However, stakeholders we interviewed noted caveats to successful stakeholder engagement that organizers must consider when implementing a systems-mapping project. Not only should organizers make targeted efforts to ensure diverse representation, but they should also consider how participation in such a project may require substantial time and energy from community members and can create emotional burden for stakeholders, especially when sensitive topics associated with ACEs come up, such as abuse. These findings are in accordance with current trauma-informed research on participant burden [60, 61]. Several participants expressed that while they valued the insight they gained about their own trauma and the trauma in their community, it was still difficult at times to share their experiences. Therefore, it is crucial that organizers of systems-mapping projects addressing trauma create an open and inclusive environment built on shared goals, understanding, compassion, and empathy.

### Limitations and implications for future research

This study summarized the reactions of 16 people to a process that engaged over 400 community stakeholders in total. Although respondents were demographically diverse and represented different sectors of the community, some participant opinions may not be represented. In particular, participants who did not value the process may have removed themselves from ROI communications and not been aware of, or interested in, contributing to this study. Additionally, participant recall bias may have impacted participant responses, given that the systems-mapping project took place in 2017–2018 and the interviews in 2020–2021.

Despite these limitations, this study highlights many promising benefits of the systems mapping process engaging members of a community in a rural, majority African-American southern county in the United States. Future research should explore how systems thinking can be leveraged to address ACEs within different community contexts and with various groups of stakeholders, as well as more downstream effects of a systems-mapping project.

## Supporting information

**S1 Table. Demographics of all stakeholders who directly participated in the systems mapping project.**
(DOCX)

**S2 Table. Interview guides.**
(DOCX)

**S3 Table. List of themes and additional illustrative quotes.**
(DOCX)

## Author Contributions

**Conceptualization:** Jared Bishop, Leigh McGill, Luke Valmadrid, Shelley Golden, Dane Emmerling, Seth Saeugling.

**Data curation:** Thi Hoang Vu, Jared Bishop, Leigh McGill, Luke Valmadrid.

**Formal analysis:** Thi Hoang Vu, Jared Bishop, Leigh McGill, Luke Valmadrid.

**Funding acquisition:** Seth Saeugling.

**Investigation:** Thi Hoang Vu, Jared Bishop, Leigh McGill, Luke Valmadrid.

**Methodology:** Thi Hoang Vu, Jared Bishop, Leigh McGill, Luke Valmadrid, Shelley Golden, Dane Emmerling, Seth Saeugling.

**Project administration:** Seth Saeugling.

**Resources:** Shelley Golden, Dane Emmerling, Seth Saeugling.

**Software:** Thi Hoang Vu, Jared Bishop, Leigh McGill, Luke Valmadrid.

**Supervision:** Shelley Golden, Dane Emmerling, Seth Saeugling.

**Validation:** Thi Hoang Vu, Jared Bishop, Leigh McGill, Luke Valmadrid, Seth Saeugling.

**Visualization:** Seth Saeugling.

**Writing – original draft:** Thi Hoang Vu.

**Writing – review & editing:** Thi Hoang Vu, Jared Bishop, Leigh McGill, Luke Valmadrid, Shelley Golden, Dane Emmerling, Seth Saeugling.

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
