## [Decision Letter · Decision Letter 0]

6 Jun 2022

PONE-D-22-02363

Using Systems-Mapping to Address Adverse Childhood Experiences (ACEs) and Trauma: A Qualitative Study of Stakeholder Experiences

PLOS ONE

Dear Dr. Vu,

Thank you for submitting your manuscript to PLOS ONE. After careful consideration, we feel that it has merit but does not fully meet PLOS ONE’s publication criteria as it currently stands. Therefore, we invite you to submit a revised version of the manuscript that addresses the points raised during the review process.

The reviewers have highlighted important strengths of this paper, but have also identified main points that require revisions at the conceptual, methodological and overall explanation level. 

We look forward to receiving your revised manuscript.

Kind regards,

Monica Wendel

Academic Editor

PLOS ONE

**Journal requirements:**

2. Please note that in order to use the direct billing option the corresponding author must be affiliated with the chosen institute. Please either amend your manuscript to change the affiliation or corresponding author, or email us at plosone@plos.org with a request to remove this option.

3. We note that Figure 1 includes an image of a [patient / participant / in the study].

**Additional Editor Comments:**

The peer review is complete, and they recognize the valuable contribution of your manuscript to the current literature. The reviewers provided several critical suggestions to strengthen the manuscript for publication.

Reviewers' comments:

Reviewer's Responses to Questions

**Comments to the Author**

1. Is the manuscript technically sound, and do the data support the conclusions?

Reviewer #1: Yes

Reviewer #2: Yes

2. Has the statistical analysis been performed appropriately and rigorously? 

Reviewer #1: Yes

Reviewer #2: Yes

3. Have the authors made all data underlying the findings in their manuscript fully available?

Reviewer #1: Yes

Reviewer #2: Yes

4. Is the manuscript presented in an intelligible fashion and written in standard English?

Reviewer #1: Yes

Reviewer #2: Yes

5. Review Comments to the Author

Reviewer #1: Using Systems Mapping to Address Adverse Childhood Experiences (ACEs) and Trauma

SM is useful in understanding and addressing ACES

Introduction:

-Describe what “1 ACES” vs “3 ACES” means. How are ACEs calculated?

-line 63 pg 3: change “attributable” to “associated with” & move the sentences about healthcare costs to the next paragraph after you talk about the kind of diseases/illnesses that ACEs contribute to

-Elaborate on current individual level interventions and the current gap

-Systems thinking is not a “critical component” of CBPR. CBPR can include systems thinking, but it can also be used to address individual level factors. I would argue that systems thinking, which is transdisciplinary in nature, requires community-based participation (in public health) but CBRP does not necessarily require systems thinking.

-The systems field typically refers to systems mapping in more specific techniques. In practice, it can be called system mapping. I suggest redefining systems mapping as community based systems dynamics or causal loop diagramming because this paper is for a technical audience, not community partners.

-Define systems thinking and cite it and systems mapping.

-I don’t know if I agree with the statement on lines 93-95. Do you have a citation?

-The sentence on line 99-100 needs to be reworded and less specific. Describe the overall type of NP and rename this section “Systems Mapping Process” not “project”

-Add more details to mapping process.

Methods:

-Move section about participants to methods & more detail about the participants. Where did they come from? Incentives, recruitment strategy, response rate etc. Was 16 the total # of people recruited?

-Describe study design (qualitative) first then talk about investigators. How many investigators?

-Were interview guides developed based on past literature? Why did you ask the questions you did? What was the overall research /evaluation question being addressed?

- I suggest using COREQ or SRSR guidelines for qualitative research to describe the methods. Look up the Equator Network to find the reporting guidelines.

-How many coders?

-What was inter-rater reliability? This is a statistic that is calculated.

Discussion:

-Based on your first finding, do you think that the SMP was an intervention within itself? It created a shared understanding and individual awareness based on your results, so it could have psychological benefits among others? I think this would be an interesting thing to add to the discussion.

-Discuss the actions resulting from SM in more detail. I think that this study is unique because it used SM to make changes. A lot of research uses SM to conceptualize issues, but doesn’t take the next steps.

Reviewer #2: Kia ora, thank you for the opportunity to read this interesting manuscript. It is a qualitative study consisting of interviews of participants from a larger study that used systems science to explore ways to address trauma related to ACES. The participants in this study were asked about their experience in the larger study, to help describe and understand the usefulness and application of systems science in the ACES subject matter. The interview analysis process was robust and thorough.

Overall this is a helpful piece of work for people to read, who may decide to use this methodology, in the fields of ACES/ Child protection/Child maltreatment.

There are a number of suggestions I have to improve this manuscript:

1. The description of the relationship of this study to the overarching ROI group and original systems science research.

a) The aim of this study needs to be much clearer at the end of the introduction. The lines 95-97 I think are the aims of this study but it would be helpful if they were framed more from the perspective of "the aim of this study was to...."

b) The aim, questions and description of the overarching systems science work are not clear and the powerpoint referenced - reference 26 cannot be accessed to understand this study better. This description needs to much more clearly state what the overall aim of the systems science research was, as well as any sub-objectives/research questions.

c) The interaction between these two studies (the overall systems science SMP project), and this sub project of analysing the process and understanding the impact, needs to be better described. At the moment it doesn't seem to work well to have the description of the ROI systems mapping project at the end of the introduction (lines 98-124). The introduction should end with the aim of the project being described in this manuscript. The detailed description of the SMP project may be better in the methods section under a section around context.

2. Purpose

Line 129 The wording around purpose should be much more including "The purpose of this study was to....."

3. Themes

It would be helpful to have a summary list of the themes before starting straight into each one. E.g. The analysis resulted in three major themes including.....

4. Background literature

At the moment there is not much in the introduction or discussion about the use of systems thinking in ACES/child protection/Child maltreatment research. It would be helpful in the background to have more about how this (or similar approaches) has been used elsewhere in this subject matter. Then in the discussion how the results, strengths and limitations within this study relate to previous studies using this methodology.

5. Minor comments

a) line 119 needs an 'of' in between 'development' and 'several'

6. PLOS authors have the option to publish the peer review history of their article (what does this mean?). If published, this will include your full peer review and any attached files.

Reviewer #1: No

Reviewer #2: No

---

## [Author Response · Author response to Decision Letter 0]

23 Jun 2022

Comments from Reviewer 1

1. Comment: Describe what “1 ACES” vs “3 ACES” means. How are ACEs calculated?

Response: We agree with this change. A clearer definition of how ACEs were calculated by the BRFSS has been included in lines 52-57.

2. Comment: line 63 pg 3: change “attributable” to “associated with” & move the sentences about healthcare costs to the next paragraph after you talk about the kind of diseases/illnesses that ACEs contribute to

Response: We agree with this change. “Attributable to” has been changed to “associated with” on line 70. The section about healthcare costs has been moved to lines 69-72 after the sentences about diseases.

3. Comment: Elaborate on current individual level interventions and the current gap

Response: We agree with this change. Lines 78-86 now highlight individual level interventions and the current gaps. 

4. Comment: Systems thinking is not a “critical component” of CBPR. CBPR can include systems thinking, but it can also be used to address individual level factors. I would argue that systems thinking, which is transdisciplinary in nature, requires community-based participation (in public health) but CBRP does not necessarily require systems thinking.

Response: We agree with this suggestion. This phrase has been removed.

5. Comment: The systems field typically refers to systems mapping in more specific techniques. In practice, it can be called system mapping. I suggest redefining systems mapping as community-based systems dynamics or causal loop diagramming because this paper is for a technical audience, not community partners.

Response: We agree with this suggestion. Lines 94-97 now refer to a systems map as the more common term and causal loop diagramming as the specific technique. 

6. Comment: Define systems thinking and cite it and systems mapping.

Response: We agree with this change. Lines 89-101 now provide definitions of systems thinking and systems-mapping with citations. 

7. Comment: I don’t know if I agree with the statement on lines 93-95. Do you have a citation?

Response: A citation has been provided on line 115.

8. Comment: The sentence on line 99-100 needs to be reworded and less specific. Describe the overall type of NP and rename this section “Systems Mapping Process” not “project”

Response: We agree with this change. The sentence has been reworded on line 131 and “Project” has been changed to “Process” on line 130.

9. Comment: Add more details to mapping process.

Response: We agree with this change. Lines 142-176 includes detailed steps of the mapping process.

10. Comment: Move section about participants to methods & more detail about the participants. Where did they come from? Incentives, recruitment strategy, response rate etc. Was 16 the total # of people recruited?

Response: Thanks for this suggestion. 

1) The authorship team considered the advantages and challenges of moving the participants section to methods and ultimately decided we prefer to leave it as is, following the examples of previously published qualitative papers in PLoS One where the participant information was included in the results section (Lasseter et al., 2018; Marcus-Varwijk et al., 2019; Nardi et al., 2020). If the editor feels strongly that we should reconsider our decision we are open to discussing it. 

2) We have added more details about the participants in lines 243-245. 

3) Incentives can be found on line 212. Sampling and recruitment information, and response rate (total reach of 200 individuals) are listed in lines 193-198. 

11. Comment: Describe study design (qualitative) first then talk about investigators. How many investigators?

Response: We agree with this suggestion and have moved study design to the beginning of the Results section on lines 181-191. We have also clarified that the investigators include: graduate students TV, JB, LM, LV, and DE in line 203; ROI staff SS in line 202; and faculty SG in line 203.

12. Comment: Were interview guides developed based on past literature? Why did you ask the questions you did? What was the overall research /evaluation question being addressed?

Response: We agree with this suggestion. This information has been added to lines 182-185, and 203-208.

13. Comment: I suggest using COREQ or SRSR guidelines for qualitative research to describe the methods. Look up the Equator Network to find the reporting guidelines.

Response: We agree with this suggestion. We have reformatted the Methods section to address the COREQ guidelines and provided a citation on lines 190-191.

14. Comment: How many coders?

Response: There were 4 coders (line 220) working in two teams of 2 (line 226)

15. Comment: What was inter-rater reliability? This is a statistic that is calculated.

Response: We did not calculate an inter-rater reliability statistic because the coding team met to discuss each code application for each transcript. Thus, each transcript was verified for coding accuracy and agreement by at least 2 coders. Additionally, discrepancies were also brought to the entire research team (TV, JB, LM, LV, DE, SG, and SS) for group consensus. Our Methods reporting are in line with previously published PLoS One qualitative papers where the inter-rater reliability statistic was also not presented ((Lasseter et al., 2018; Marcus-Varwijk et al., 2019; Nardi et al., 2020).

16. Comment: Based on your first finding, do you think that the SMP was an intervention within itself? It created a shared understanding and individual awareness based on your results, so it could have psychological benefits among others? I think this would be an interesting thing to add to the discussion.

Response: We agree that this is an intriguing discussion point that should be added. An additional paragraph addressing this point has been added in lines 480-495.

17. Comment: Discuss the actions resulting from SM in more detail. I think that this study is unique because it used SM to make changes. A lot of research uses SM to conceptualize issues, but doesn’t take the next steps.

Response: We agree with this suggestion. Additional details about actions resulting from the SMP has been added in lines 462-270.

Comments from Reviewer 2

18. Comment: The aim of this study needs to be much clearer at the end of the introduction. The lines 95-97 I think are the aims of this study but it would be helpful if they were framed more from the perspective of "the aim of this study was to...."

Response: We agree with this suggestion. We have reframed the aims in lines 125-128 for clarity.

19. Comment: The aim, questions and description of the overarching systems science work are not clear and the powerpoint referenced - reference 26 cannot be accessed to understand this study better. This description needs to much more clearly state what the overall aim of the systems science research was, as well as any sub-objectives/research questions.

Response: We agree with this suggestion. We have added lines 111-176 to describe how the overarching systems-science work, the SMP, and the qualitative interviews relate to and build off of each other. The link to the PowerPoint reference has been fixed and is now accessible. 

20. Comment: The interaction between these two studies (the overall systems science SMP project), and this sub project of analysing the process and understanding the impact, needs to be better described. At the moment it doesn't seem to work well to have the description of the ROI systems mapping project at the end of the introduction (lines 98-124). The introduction should end with the aim of the project being described in this manuscript. The detailed description of the SMP project may be better in the methods section under a section around context.

Response: We agree with this suggestion. In addition to the changes made in point #19, we have moved the description of the SMP into the Methods section labeled “Context: The ROI Systems-Mapping Process”

21. Comment: Line 129 The wording around purpose should be much more including "The purpose of this study was to....."

Response: After reorganizing and rewording the last paragraph of the introduction, we have removed this sentence.

22. Comment: It would be helpful to have a summary list of the themes before starting straight into each one. E.g. The analysis resulted in three major themes including.....

Response: We agree with this suggestion. An additional section has been added under Results describing a summary of the themes in lines 262-269.

23. Comment: At the moment there is not much in the introduction or discussion about the use of systems thinking in ACES/child protection/Child maltreatment research. It would be helpful in the background to have more about how this (or similar approaches) has been used elsewhere in this subject matter. Then in the discussion how the results, strengths and limitations within this study relate to previous studies using this methodology.

Response: We agree with this suggestion. We have added additional information in the introduction about the use of systems-thinking in ACEs research in lines 101-110, as well as relating our study to this previously published research in the discussion in lines 439-443.

24. Comment: line 119 needs an 'of' in between 'development' and 'several'

Response: This change has been made in line 171.

---

## [Decision Letter · Decision Letter 1]

18 Jul 2022

PONE-D-22-02363R1Using Systems-Mapping to Address Adverse Childhood Experiences (ACEs) and Trauma: A Qualitative Study of Stakeholder ExperiencesPLOS ONE

Dear Dr. Vu,

Thank you for submitting your manuscript to PLOS ONE. After careful consideration, we feel that it has merit but does not fully meet PLOS ONE’s publication criteria as it currently stands. Therefore, we invite you to submit a revised version of the manuscript that addresses the points raised during the review process.

Thanks for the very nice revision. Please note that one of the reviewers still has some minor comments that need to be addressed before I can make a final decision on publication. 

We look forward to receiving your revised manuscript.

Kind regards,

Sara Rubinelli

Academic Editor

PLOS ONE

Journal Requirements:

Reviewers' comments:

Reviewer's Responses to Questions

**Comments to the Author**

1. If the authors have adequately addressed your comments raised in a previous round of review and you feel that this manuscript is now acceptable for publication, you may indicate that here to bypass the “Comments to the Author” section, enter your conflict of interest statement in the “Confidential to Editor” section, and submit your "Accept" recommendation.

Reviewer #2: All comments have been addressed

2. Is the manuscript technically sound, and do the data support the conclusions?

Reviewer #2: Yes

3. Has the statistical analysis been performed appropriately and rigorously? 

Reviewer #2: Yes

4. Have the authors made all data underlying the findings in their manuscript fully available?

Reviewer #2: Yes

5. Is the manuscript presented in an intelligible fashion and written in standard English?

Reviewer #2: Yes

6. Review Comments to the Author

Reviewer #2: Thank you for the changes made in response to review.

Just a couple of minor edits needed:

Line 157 “LLC” is this an acronym – if so expand.

Lines 182-183 “: 1) What were the 183 perceived individual, interpersonal, and societal level impacts the SMP?”

Should there be an ‘of’ between ‘impacts’ and ‘the’

And a couple of general comments:

Names were in manuscript - was this supposed to be a double blind review. If so in future please anonymise your manuscript.

Not sure why original manuscript was submitted in the resubmission. It was confusing initially.

7. PLOS authors have the option to publish the peer review history of their article (what does this mean?). If published, this will include your full peer review and any attached files.

Reviewer #2: No

---

## [Author Response · Author response to Decision Letter 1]

23 Jul 2022

Comments from Reviewer 2:

1. Comment: Line 157 “LLC” is this an acronym – if so expand.

Response: We agree with this suggestion. We have removed the term “LLC” on line 157 because the sentence following “Engaging Inquiry” already describes what the organization is. 

2. Comment: Lines 182-183 “: 1) What were the 183 perceived individual, interpersonal, and societal level impacts the SMP?”

Should there be an ‘of’ between ‘impacts’ and ‘the’

Response: We apologize for the typo. We added an “of” on line 183. 

3. Comment: Names were in manuscript - was this supposed to be a double blind review. If so in future please anonymise your manuscript.

Response: We were informed by the journal that all authors’ names must be listed in the title page. 

4. Comment: Not sure why original manuscript was submitted in the resubmission. It was confusing initially.

Response: We removed all former manuscript versions from the submission portal and are only submitting the most recent revised versions in this round. We apologize for the confusion.

---

## [Editor Report · Decision Letter 2]

8 Aug 2022

Using Systems-Mapping to Address Adverse Childhood Experiences (ACEs) and Trauma: A Qualitative Study of Stakeholder Experiences

PONE-D-22-02363R2

Dear Dr. Vu,

We’re pleased to inform you that your manuscript has been judged scientifically suitable for publication and will be formally accepted for publication once it meets all outstanding technical requirements.

Kind regards,

Sara Rubinelli

Academic Editor

PLOS ONE
---

## [Editor Report · Acceptance letter]

10 Aug 2022

PONE-D-22-02363R2 

Using Systems-Mapping to Address Adverse Childhood Experiences (ACEs) and Trauma: A Qualitative Study of Stakeholder Experiences

Dear Dr. Vu:

I'm pleased to inform you that your manuscript has been deemed suitable for publication in PLOS ONE. Congratulations! Your manuscript is now with our production department. 

Kind regards, 

on behalf of

Dr. Sara Rubinelli 

Academic Editor

PLOS ONE